# Triglyceride-Glucose Index in Non-Diabetic, Non-Obese Patients with Obstructive Sleep Apnoea

**DOI:** 10.3390/jcm10091932

**Published:** 2021-04-29

**Authors:** Andras Bikov, Stefan M. Frent, Martina Meszaros, Laszlo Kunos, Alexander G. Mathioudakis, Alina Gabriela Negru, Laura Gaita, Stefan Mihaicuta

**Affiliations:** 1North West Lung Centre, Wythenshawe Hospital, Manchester University NHS Foundation Trust, Manchester M23 9LT, UK; andras.bikov@gmail.com (A.B.); a.mathioudakis@nhs.net (A.G.M.); 2Division of Infection, Immunity & Respiratory Medicine, University of Manchester, Manchester M23 9LT, UK; 3Center for Research and Innovation in Precision Medicine of Respiratory Diseases, Department of Pulmonology, “Victor Babes” University of Medicine and Pharmacy Timisoara, Eftimie Murgu Sq. no. 2, 300041 Timisoara, Romania; stefan.mihaicuta@umft.ro; 4Department of Pulmonology, Semmelweis University, 1085 Budapest, Hungary; martina.meszaros1015@gmail.com (M.M.); laszlo.kunos@gmail.com (L.K.); 5Department of Cardiology, “Victor Babes” University of Medicine and Pharmacy Timisoara, Eftimie Murgu Sq. no. 2, 300041 Timisoara, Romania; eivanica@yahoo.com; 6Institute of Cardiovascular Diseases Timisoara, 300310 Timisoara, Romania; 7Department of Internal Medicine II, “Victor Babes” University of Medicine and Pharmacy Timisoara, Eftimie Murgu Sq. no. 2, 300041 Timisoara, Romania; laura_gaita@yahoo.com

**Keywords:** obstructive sleep apnea, triglyceride-glucose index, insulin resistance, diabetes, obesity

## Abstract

Obstructive sleep apnoea (OSA) is associated with increased insulin resistance. Triglyceride-glucose index (TyG) is a simple marker of insulin resistance; however, it has been investigated only by two studies in OSA. The aim of this study was to evaluate TyG in non-diabetic, non-obese patients with OSA. A total of 132 patients with OSA and 49 non-OSA control subjects were included. Following a diagnostic sleep test, fasting blood was taken for the analysis of the lipid profile and glucose concentrations. TyG was calculated as ln(triglyceride [mg/dL] × glucose [mg/dL]/2). Comparison analyses between OSA and control groups were adjusted for age, gender, body mass index (BMI) and smoking. TyG was higher in men (*p* < 0.01) and in ever-smokers (*p* = 0.02) and it was related to BMI (ρ = 0.33), cigarette pack-years (ρ = 0.17), apnoea–hypopnoea index (ρ = 0.38), oxygen desaturation index (ρ = 0.40), percentage of total sleep time spent with oxygen saturation below 90% (ρ = 0.34), and minimal oxygen saturation (ρ = −0.29; all *p* < 0.05). TyG values were significantly higher in OSA (*p* = 0.02) following adjustment for covariates. OSA is independently associated with higher TyG values which are related to disease severity in non-obese, non-diabetic subjects. However, the value of TyG in clinical practice should be evaluated in follow-up studies in patients with OSA.

## 1. Introduction

Obstructive sleep apnoea (OSA) is a common disease characterised by a repetitive, partial or total collapse of the upper airways during sleep resulting in intermittent hypoxaemia and sleep fragmentation. It is a known risk factor for the development of cardiovascular and metabolic diseases, such as type 2 diabetes mellitus (T2DM) [1].

Insulin resistance is a clinical condition in which insulin exerts a biological effect lower than expected and, in most cases, it is associated with metabolic abnormalities such as hyperglycaemia and hypertriglyceridemia [2]. In insulin resistance, adipocytes release free fatty acids (FFA) which are taken up by the liver and form triglyceride (TG)-rich, very-low-density lipoprotein (VLDL) particles. These particles exchange lipids with low-density lipoprotein (LDL) and high-density lipoprotein (HDL) particles. Triglyceride rich HDL particles are removed rapidly by the kidneys; hence they cannot participate in reverse cholesterol transport [2]. Besides, insulin resistance and hyperglycaemia independently contribute to endothelial dysfunction, vascular smooth muscle proliferation and inflammation [3,4,5]. Hence, it is not surprising that insulin resistance is strongly associated with atherosclerosis [6] and more particularly, coronary artery disease [7].

Chronic intermittent hypoxaemia is considered the most important mechanism leading to insulin resistance in obstructive sleep apnoea [8,9]. More particularly, intermittent hypoxaemia downregulates the insulin receptor in adipocytes and skeletal muscles [10] which process is augmented by the local inflammation in the adipose tissue [11]. Other mechanisms include β cell dysfunction, sympathetic bursts and hormonal changes which induce hyperglycaemia and impaired tissue responsiveness to insulin [12]. Supporting this, there is an independent association between OSA and components of the metabolic syndrome, particularly, insulin resistance and abnormal lipid metabolism [13].

The homeostasis model assessment of insulin resistance (HOMA-IR) value is often used to evaluate insulin resistance in OSA [14]. The model relies on the measurement of fasting insulin levels which is not always available in every laboratory, besides the assay-dependent variability of serum insulin between commercially available assays [15]. The triglyceride glucose index (TyG) is a novel marker of insulin resistance that requires fasting triglyceride and plasma glucose measurements and it was shown to correlate with other markers of insulin resistance, such as HOMA-IR and the hyperinsulinemic-euglycemic clamp [16]. Only two studies have evaluated TyG in OSA so far and reported higher levels in OSA than in controls, as well as a significant relationship with disease severity [17,18].

Obesity is a significant factor contributing to OSA risk [19] and is also essential in the development of insulin resistance. On one hand, increased FFA production in obesity can inhibit insulin signalling contributing to insulin resistance [20]. On the other hand, adipokines secreted by the adipose tissue also reduce the effect of insulin and lead to increased hepatic glucose production [21]. Not surprisingly, obesity was found to be a major confounder influencing the relationship between OSA and metabolic dysfunction [22,23]. However, the previous two studies that have evaluated TyG in OSA did not exclude patients with obesity [17,18]. Therefore, the findings could be biased by this significant confounder.

We hypothesised that OSA would lead to elevated TyG value in non-obese patients without diabetes. Therefore, the aim of the current study was to evaluate TyG values in these subjects and to correlate them with markers of disease severity.

## 2. Methods

### 2.1. Study Subjects and Design

In total, 687 consecutive adult patients referred to two sleep laboratories (Budapest, Hungary and Timisoara, Romania) due to symptoms suggestive for obstructive sleep apnoea (snoring, witnessed pauses in breathing, daytime tiredness, excessive daytime sleepiness) and participating in controlled observational studies [24] were evaluated. For this analysis, we included participants with a body mass index (BMI) < 30 kg/m^2^ and those with available fasting blood glucose and triglyceride measurements. Patients diagnosed and treated for T2DM, those with fasting blood glucose > 126 mg/dL and those with acute respiratory, heart and renal failure were excluded as the later three could acutely affect the sleep study findings. Hence, 181 volunteers satisfied the inclusion and exclusion criteria and their data were available for analysis (Figure 1).

After taking a detailed medical history, participants filled out the Epworth Sleepiness Scale (ESS) questionnaire. This was followed by an inpatient sleep test (*n* = 150 polysomnography and 31 cardiorespiratory polygraphy). The following day, a fasting blood sample was collected for glucose, triglycerides (TG), total cholesterol, LDL-cholesterol (LDL-C), and HDL-cholesterol (HDL-C) measurements. TyG was calculated as ln(TG × glucose/2), with TGs and blood glucose levels expressed in mg/dL. At the Hungarian site bloods were analysed with the AU680 Clinical Chemistry Analyzer (Beckman Coulter, UK). Triglyceride levels were estimated by the enzymatic GPO-POD method. The interassay coefficient of variation (CV%) for 0.47, 4.28 and 10.20 mmol/L were 1.76%, 1.03% and 1.46%, respectively. Glucose levels were estimated with the hexokinase method. The interassay CV% for 2.99, 6.43 and 16.31 mmol/L were 0.9%, 0.6% and 0.7%, respectively. The laboratory is not ISO15189-certified. The blood samples were analysed at the Romanian site with a COBAS INTEGRA 400 plus analyzer (Roche Diagnostics, Basel, Switzerland). TG, total cholesterol, LDL-C and HDL-C levels were estimated by the enzymatic colorimetric method with an interassay CV% of 1.6%, 0.5%, 1.3% and 1.13%, respectively. Glucose levels were assessed using the enzymatic hexokinase method with a CV% of 0.7%. The laboratory is SR EN ISO 15189:2013 certified. Comorbidities were defined by patient reports, available medical and drug charts. In case of uncertainty, patients were referred to specialist care to confirm the diagnosis of the comorbidity.

All procedures performed in our study involving human participants were in accordance with the ethical standards of the 1964 Helsinki declaration and its later amendments. The study was approved by the local Ethics Committees (Semmelweis University TUKEB 30/2014 and RKEB 172/2018, and University of Medicine and Pharmacy Victor Babes Timisoara 22/2014/24.07.2019) and patients gave their informed consent before participating in the study.

### 2.2. Sleep Studies

Inpatient cardiorespiratory polygraphy and polysomnography were performed according to the American Academy of Sleep Medicine (AASM) recommendations [25]. Sleep stages, movements and cardiopulmonary events were scored manually according to the AASM guidelines [26]. Apnoea was defined as ≥ 90% drop in the nasal flow lasting for ≥ 10 s. Hypopnoea was defined as ≥ 30% drop in the nasal flow lasting for ≥ 10 s which is associated with either ≥ 3% drop in the oxygen saturation (for both polygraphy and polysomnography) or arousal (for polysomnography). We recorded total sleep time (TST), sleep period time (SPT), percentage of rapid eye movement sleep (REM%) and minimal oxygen saturation (MinSatO_2_) and calculated sleep efficiency (Sleep%, TST/SPT), apnoea–hypopnoea index (AHI), oxygen desaturation index (ODI), and the percentage of total sleep time spent with saturation below 90% (TST90%).

The diagnosis of OSA was based on AHI ≥ 5/h and suggestive symptoms according to the International Classification of Sleep Disorders (Third Edition) criteria. The OSA group was divided into mild (AHI 5–14.9/h), moderate (AHI 15–29.9/h) and severe (AHI ≥ 30/h) subgroups. The control group comprised patients who were referred due to symptoms suggestive for OSA, but had AHI < 5/h on the sleep test.

### 2.3. Statistical Analysis

JASP 0.14 (JASP Team, University of Amsterdam, The Netherlands) was used for statistical analysis. The normality of the data was assessed with the Shapiro–Wilk test. The OSA and control groups were compared with *t*-test, Mann–Whitney, Fisher and Chi-square tests with Yates’ correction. The relationships between TyG and clinical as well as sleep parameters were studied with Spearman’s test. Sensitivity analyses for these correlations in men and women as well as in lean (BMI < 25 kg/m^2^) and overweight (BMI 25–29.9 kg/m^2^) subjects were also performed. The relationships between TyG and OSA as well as TyG and OSA severity were investigated with the analysis of covariance (ANCOVA) test adjusted for age, gender, smoking status and BMI followed by the Tukey’s post hoc test to compare various severity subgroups. Data are expressed as mean ± standard deviation for parametric or median/interquartile range for nonparametric variables. A *p*-value < 0.05 was considered significant.

The primary aim of this study was to compare TyG values between the OSA and control groups. The unadjusted difference (effect size) between patients with OSA and controls was 0.74 in a previous case–control study [17]. As in this study we investigated a selected non-obese, non-diabetic population, we hypothesised that the difference between the two groups would be at least 50% smaller. To detect this (0.37) effect size in the adjusted model with a power of 0.80 and α error probability of 0.05 we had to recruit 128 subjects [27]. As data from more subjects were eligible to analyse, we decided to include those results, so that we could increase the power of the study.

## 3. Results

### 3.1. Demographics and Clinical Characteristics of the OSA and Control Groups

A total of 132 non-obese, non-diabetic patients were diagnosed with OSA (43 mild, 39 moderate, 50 severe). Patients with OSA were older, had higher BMI, cigarette pack-years, TG and glucose levels, higher AHI, ODI and TST90%, longer SPT and lower HDL-C levels and MinSatO_2_ compared to the controls (all *p* < 0.05). Besides, the OSA group had a higher proportion of ever-smokers, patients with hypertension, cardiovascular and cerebrovascular diseases (Table 1).

### 3.2. Relationship between Triglyceride-Glucose Index and Demographics as Well as Sleep and Clinical Characteristics

TyG was higher in men (*n* = 109, 8.73 ± 0.50 vs. 8.46 ± 0.44, *p* < 0.01) and in ever-smokers (*n* = 53, 8.76 ± 0.49 vs. 8.57 ± 0.49, *p* = 0.02); however, the latter became insignificant following adjustment for age, BMI and gender (*p* = 0.50). Interestingly, there was no difference in TyG between patients with and without hypertension (*p* = 0.13), cardiovascular and cerebrovascular disease (*p* = 0.16) or arrhythmia (*p* = 0.32).

There was a significant relationship between TyG and BMI (ρ = 0.33), cigarette pack-years (ρ = 0.17), AHI (ρ = 0.38), ODI (ρ = 0.40), TST90% (ρ = 0.34) and MinSatO_2_ (ρ = −0.29, all *p* < 0.05). In contrast, no relationship was present between TyG and ESS, TST, SPT, Sleep% or REM% (all *p* > 0.05, Appendix A).

Analysing women separately (*n* = 72), significant relationships between TyG and age (ρ = 0.33), BMI (ρ = 0.46), AHI (ρ = 0.44), ODI (ρ = 0.49), TST90% (ρ = 0.45) and MinSatO_2_ (ρ = −0.42, all *p* < 0.05) were noted. Additionally, women with hypertension (8.66 ± 0.43 vs. 8.30 ± 0.38, *p* < 0.001) as well as cerebrovascular and cardiovascular disease (8.77 ± 0.43 vs. 8.41 ± 0.42, *p* = 0.01) had higher TyG values. In men (*n* = 109), significant correlations were presented between TyG and AHI (ρ = 0.22), ODI (ρ = 0.24) and TST90% (ρ = 0.22). No other relationships with sleep characteristics or comorbidities were present in either gender (all *p* > 0.05).

In lean subjects (*n* = 64), TyG was related to BMI (ρ = 0.27), AHI (ρ = 0.34), ODI (ρ = 0.43), TST90% (ρ = 0.32) and MinSatO_2_ (ρ = −0.38, all *p* < 0.05). Lean patients with hypertension had higher TyG values compared to those without hypertension (8.62 ± 0.45 vs. 8.36 ± 0.45, *p* = 0.04). In overweight participants (*n* = 117), significant relationships between TyG and AHI (ρ = 0.23), ODI (ρ = 0.24) and TST90% (ρ = 0.21) were noted. There was no relationship between TyG and BMI, comorbidities, or any other variables.

### 3.3. Triglyceride-Glucose Index in OSA vs. Controls

There was a significant difference in TyG between patients with OSA (8.72 ± 0.49) and controls (8.37 ± 0.40, *p* < 0.01). The difference remained significant after adjusting for age, gender, smoking status and BMI (*p* = 0.03, Figure 2).

There was a significant relationship between worsening OSA severity and progressively increasing TyG values (adjusted *p* = 0.04, Figure 3). Besides, a significant difference was noticed between the control (8.37 ± 0.40) and severe OSA (8.84 ± 0.49) groups (post-hoc Tukey test, *p* = 0.03). No difference in TyG values between the control and mild OSA (*p* = 0.31) or moderate OSA (*p* = 0.53) groups was observed. There was no difference between the mild and moderate (*p* = 0.99), mild and severe (*p* = 0.54) or between the moderate and severe (*p* = 0.36) groups.

## 4. Discussion

In this study, we investigated the triglyceride-glucose index in non-diabetic, non-obese subjects with obstructive sleep apnoea. We report that this marker is independently associated with OSA and relates to disease severity.

Only three studies examined TyG in OSA so far [17,18,28], but only two used objective sleep tests [17,18]. Zou et al. studied 4703 Chinese participants and found that TyG is associated with OSA and relates to disease severity [18]. Kang et al. concluded similar results investigating 180 Korean participants [17]. Most importantly, neither of these studies considered obesity and diabetes as confounders. In addition, the authors did not evaluate the relationship between TyG and other sleep parameters than AHI. Moreover, as the relationship between OSA and metabolic consequences is influenced by genetic factors [29] as well as regional differences (cuisine, access to healthcare systems, etc.) [24], a study performed on East Asian subjects cannot necessarily be extrapolated to other populations. Of note, the current study is the first to investigate TyG in OSA in a Caucasian population. Bianchi et al. evaluated TyG in patients with abdominal aortic aneurysm and reported higher levels in those with high OSA risk based on the Berlin questionnaire; however, no objective test to confirm OSA was performed [28].

In line with the literature [8], chronic intermittent hypoxaemia was the predominant mechanism leading to augmented insulin resistance in patients with OSA in our study. Chronic intermittent hypoxaemia induces the expression of hypoxia-inducible factor 1 (HIF-1) [30,31] which is a key factor in the regulation of metabolism [32]. However, sleep fragmentation, sleep restriction or sleep deprivation could also contribute to insulin resistance [33]. Although both previous studies used polysomnography to evaluate OSA [17,18], the relationship between markers of sleep quality and TyG has not been investigated before. In the current study, we report no correlation, which is in line with our previous report that the lipid profile does not strongly relate to markers of sleep quality in OSA [34]. Of note, polysomnography has been performed only in a proportion of the subjects, therefore, the results of this analysis need to be interpreted carefully. In addition, medications, such as statins or corticosteroids may affect sleep quality [35]. We did not analyse the effect of medications on the correlations between TyG and sleep quality indices and this could have also contributed to the lack of findings.

Similarly to the previous study [18], male patients with OSA had higher TyG values. The intergender differences in metabolism, including insulin resistance are well known and are attributed to genetic, epigenetic, and hormonal factors [36]. Male gender also elevates the risk for OSA [19] and the primary analyses were adjusted for gender to exclude this effect. In addition, relationships between TyG and OSA severity were performed in men and women separately. In both settings, AHI was related to TyG values irrespective of gender.

Although obesity was excluded, a significant relationship was observed between BMI and TyG values. This correlation is not surprising and is more prominent in the high BMI range [37]. Therefore, to mitigate this effect, the primary analyses were also adjusted for BMI and correlation analyses were performed separately in lean and overweight subjects. The relationship between TyG and OSA severity was demonstrated following adjustment for BMI and in both lean and overweight subjects separately. However, TyG values correlated with BMI even in lean subjects, reinforcing that studies investigating OSA and metabolic parameters should always consider BMI as a continuous confounding variable. Interestingly, there was no correlation between TyG and BMI in overweight volunteers. The exact reason is not clear, however, the lack of relationship has to be interpreted with caution; first, because this raw analysis was not adjusted for covariates; second, because this was performed in a limited number of subjects.

Age is a significant predictor for OSA [19], therefore our analyses were adjusted for this cofactor. However, there was no relationship between age and TyG, therefore, the age-adjusted ANCOVA model has to be interpreted carefully, as the criteria for the linearity between the dependent variable and the covariant was not met.

Around 20–25% of patients with OSA are diagnosed with cardiovascular or cerebrovascular disease [24], and 80% have coronary artery plaques [38]. Previous studies have established TyG as a marker to predict clinical [39,40] and subclinical atherosclerotic disease [41] and its progression [42]. Moreover, Irace et al. reported that TyG better predicted atherosclerosis than the HOMA-IR [43]. Hypertension is a risk factor for cardiovascular disease. In line with the previous studies [44,45], we noted a trend for a relationship between TyG and hypertension which was significant in women only. Interestingly, Lambrinoudaki et al. concluded that TyG could be particularly useful in predicting atherosclerosis in lean and overweight women [46]. We did not find a relationship between TyG and self-reported atherosclerotic cardiovascular disease, only in a subgroup of women. However, in this study, we did not perform objective tests to confirm or exclude coronary or cerebrovascular disease. Nevertheless, to assess if TyG is useful to predict the development of atherosclerotic cardiovascular disease in OSA, follow up studies are warranted. The clinical need for such biomarkers has been emphasised by the European Respiratory Society and the European Sleep Research Society [47].

This study has limitations. First, we used cardiorespiratory polygraphy rather than polysomnography in 31 cases (17%). Polygraphy is an accepted method to diagnose OSA in non-complicated cases and high OSA probability according to the AASM guidelines [48]. However, some of the respiratory events (i.e., hypopnoea associated with arousal but without desaturations) may not have been scored. Therefore, analyses with OSA severity must be interpreted carefully. Secondly, although fasting blood samples were analysed, long term diet and exercise may affect plasma glucose and triglyceride values. Unfortunately, we did not record these data in the study. Thirdly, this was a cross-sectional study. Prospective clinical trials are warranted to investigate the clinical value of TyG in OSA and most importantly, the effect of OSA treatment. We believe that our study could serve as a basis to design such trials.

## 5. Conclusions

In summary, we reported higher TyG values in non-diabetic, non-obese patients with OSA independently from age, gender and BMI. TyG, being a feasible and cheap marker for insulin resistance, should be routinely used to predict cardiovascular disease risk in patients with OSA.

## Figures and Tables

**Figure 1 jcm-10-01932-f001:**
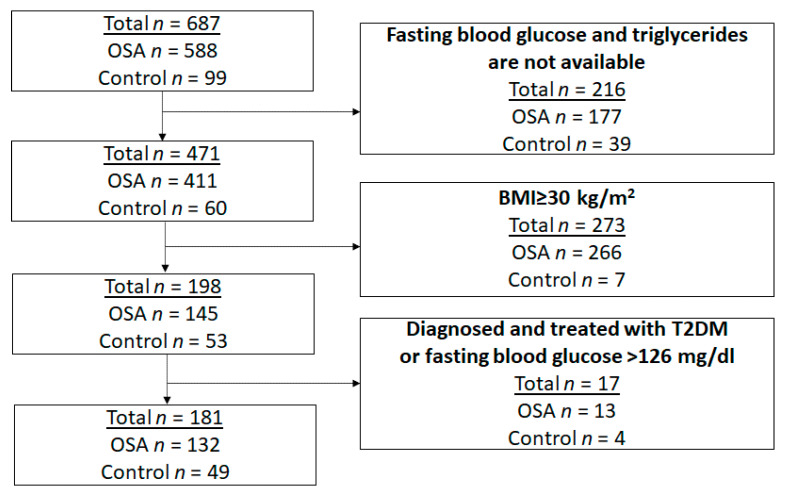
The algorithm for patient selection.

**Figure 2 jcm-10-01932-f002:**
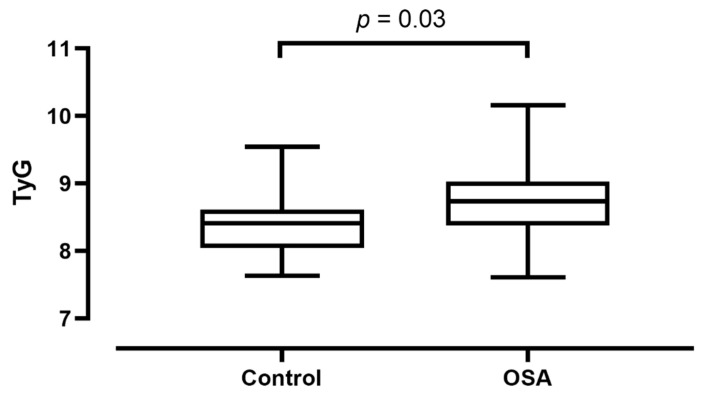
Triglyceride-glucose index (TyG) values compared between controls and patients with obstructive sleep apnoea (OSA). Median, minimum and maximum values are plotted together with the interquartile range.

**Figure 3 jcm-10-01932-f003:**
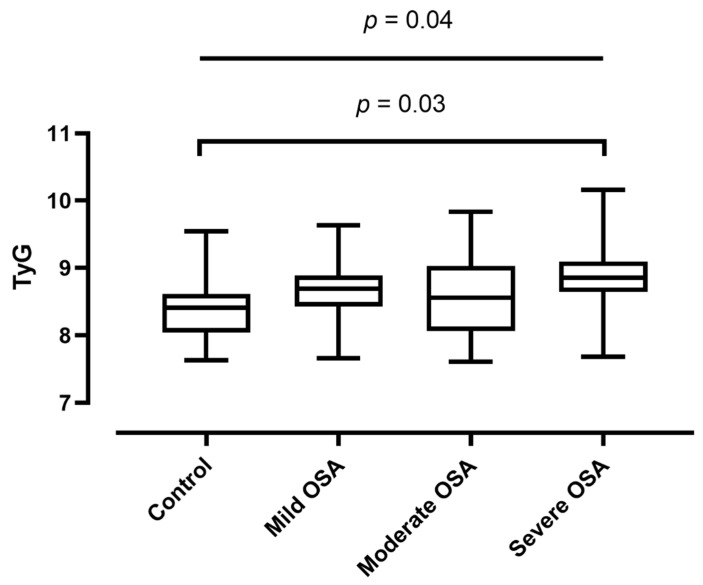
Triglyceride-glucose index (TyG) values compared between groups of increasing obstructive sleep apnoea (OSA) severity. Median, minimum and maximum values are plotted together with the interquartile range.

**Table 1 jcm-10-01932-t001:** Demographics and clinical characteristics of the OSA and control groups.

	OSA (*n* = 132)	Control (*n* = 49)	*p*
Age (years)	53/44–60/	47/29–59/	0.03
Gender (male%)	70	33	<0.01
BMI (kg/m^2^)	26.8/24.8–28.6/	23.9/20.8–26.9/	<0.01
Smoking (ever%)	39	4	<0.01
Cigarette pack years	0/0–11/	0/0–0/	<0.01
Hypertension (%)	55	35	0.03
Cardiovascular and cerebrovascular disease (%)	23	4	<0.01
Arrhythmia (%)	22	18	0.75
Total cholesterol (mmol/L)	5.0/4.3–6.1/	5.2/4.7–5.9/	0.28
HDL-C (mmol/L)	1.1/1.0–1.4/	1.6/1.4–2.0/	<0.01
LDL-C (mmol/L)	2.9/2.3–3.8/	3.0/2.6–3.7/	0.85
TG (mmol/L)	1.7/1.2–2.0/	1.2/0.9–1.5/	<0.01
Glucose (mmol/L)	4.8/4.4–5.3/	4.7/4.2–4.9/	0.02
TyG	8.72 ± 0.49	8.37 ± 0.40	<0.01
ESS	8.0/5.0–10.0/	6.0/4.0–9.0/	0.07
TST (min)	420/367–453/	388/354–431/	0.11
SPT (min)	466/430–505/	425/398–463/	<0.01
Sleep% (%)	92/82–96/	94/84–99/	0.12
REM% (%)	15/12–19/	14/10–19/	0.11
AHI (1/h)	24.1/12.2–40.7/	1.9/1.2–2.8/	<0.01
ODI (1/h)	21.4/9.9–35.4/	0.9/0.2–1.7/	<0.01
TST90% (%)	3.4/0.2–13.2/	0.0/0.0–0.1/	<0.01
MinSatO_2_ (%)	85/81–88/	91/89–93/	<0.01

AHI—apnoea–hypopnoea index, BMI—body mass index, ESS—Epworth Sleepiness Scale, HDL-C—high-density lipoprotein-cholesterol, LDL-C—low-density lipoprotein-cholesterol, MinSatO_2_—minimal oxygen saturation, ODI—oxygen desaturation index, OSA—obstructive sleep apnoea, REM%—percentage of total sleep time spent in rapid eye movement sleep, Sleep%—sleep efficiency, SPT—sleep period time, TG—triglyceride, TST—total sleep time, TST90%—percentage of total sleep time spent with oxygen saturation below 90%, TyG—triglyceride-glucose index. Data are expressed as mean ± standard deviation or median /interquartile range/.

## Data Availability

The data of the study are available upon request to the corresponding author.

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
