# Peer review of "Triglyceride-Glucose Index in Non-Diabetic, Non-Obese Patients with Obstructive Sleep Apnoea"

_jcm, 2021, doi:10.3390/jcm10091932_

Round 1
Reviewer 1 Report
General comments
The authors investigated TyG in non-obese and non-diabetic OSA patients, taking into account for many confounding factors (age, gender, BMI, smoking, CVD). They show that TyG is independently and positively associated with OSA severity, on the whole group, as well as in lean and overweight subgroups. Overall, the design is relevant, the methods and results are presented and discussed appropriately.
Nevertheless, the following points should be considered:
Major comments
1) Methods
Given that TyG, the key marker of the study, is based on biological markers, some methodological information are required for TG and FPG: the methods (e.g. hexokinase), the analyser(s) and manufacturer(s) (e.g. Cobas C701, Roche Diagnostics, Mannheim, Germany, or Architect C16000, Abbott, Chicago, US). The main analytical performances are also required (ideally those established by the local laboratory, or at least those provided in the manufacturer’ technical sheets), in particular the inter-assay variability (CV% for at least two concentration levels). If the local laboratory is ISO15189-certified, then it should be mentioned.
2) Statistics
The authors used an ANCOVA to study the relationship between TyG and OSA and OSA severity, taking account for confounders. Did the authors verify the ANCOVA-related hypotheses? Notably 1) the assumption of linearity between the response and covariates, 2) the homogeneity between regression slopes, which should be the same for each group (this hypothesis tests that there is not interaction between the result and the covariate: regression lines by group should be parallel), 3) the homogeneity of residuals’ variance for all subgroups (residuals are supposed to have a constant variance, i.e., homoscedasticity), and 4) the absence of significant aberrant values. If one or more of these assumptions are not met, then a robust ANCOVA could be assessed or, at least, these statistical limits should be mentioned as another limitation of the study.
3) Results
TyG is positively associated with BMI in men (whole?) group (P5, L165; rho=0.33), in women (L173; rho=0.46), as well as in lean patients (L179; rho=0.27). It is therefore surprising that such a correlation was not observed in patients with overweight (L184). How do the authors explain this paradoxical result?
Minor comments
1) Introduction: Complete the sentence about insulin (P2, L66): “…is not always available in every laboratory” by adding “, besides the assay-dependent variability of serum insulin between commercially available assays [PMID 28150502]”.
2) Given the many correlations, a correlogram would have been relevant, allowing a better visualization, displaying plots, regression lines, spearman rho and p-value for each bivariate correlation.
3) The figures 2 and 3 should be reassessed as box and whisker plots to show the IQR, and with a lower size of plots to show up the medians, which are currently more or less hidden.
4) There is a discrepancy between the “p=0.03” on Figure 3 (ctrl vs severe OSA) and the “p=0.04” in the text (P5, L195). P-value would be instructive, for each of these subgroups. Remove “in all subjects” form the 3.2 subtitle (P5, L162).
Reviewer 2 Report
The authors took up a very current and important topic of the triglyceride-glucose index in non-diabetic, non-obese patients with obstructive sleep apnoea. In my opinion, the manuscript in this shape requires revision. The following issues required correction:
Major:
- Authors stated that sample size calculation was performed. Its results should be provided in the manuscript.
- My main concern is the poor bibliography. The calculated minimal sample size must be provided in the full text!
b) Authors are talking about apnoea-hypopnoea index, oxygen desaturation index, percentage of total sleep time spent with oxygen saturation below 90%, and minimal oxygen saturation. In my opinion, these parameters should be discussed with the molecular confirmation of hypoxia (e.g. https://www.mp.pl/paim/issue/article/15104/, https://jcsm.aasm.org/doi/abs/10.5664/jcsm.8682)
b) what with lipid-lowering drugs and their effect on this index/sleep quality (e.g. https://www.sciencedirect.com/science/article/pii/S1087079220301234)?
Minor:
- the p values should be reported with the same accuracy
Round 2
Reviewer 1 Report
The authors responded in a very appropriate manner to each of my comments, with clarity, transparency and precision.
Reviewer 2 Report
Authors addressed all the comments. I recommend the manuscript for the publication.